# Association between HLA DNA Variants and Long-Term Response to Anti-TNF Drugs in a Spanish Pediatric Inflammatory Bowel Disease Cohort

**DOI:** 10.3390/ijms24021797

**Published:** 2023-01-16

**Authors:** Sara Salvador-Martín, Paula Zapata-Cobo, Marta Velasco, Laura M. Palomino, Susana Clemente, Oscar Segarra, Cesar Sánchez, Mar Tolín, Ana Moreno-Álvarez, Ana Fernández-Lorenzo, Begoña Pérez-Moneo, Inés Loverdos, Victor Manuel Navas López, Antonio Millán, Lorena Magallares, Ricardo Torres-Peral, Ruth García-Romero, Gemma Pujol-Muncunill, Vicente Merino-Bohorquez, Alejandro Rodríguez, Enrique Salcedo, Beatriz López-Cauce, Ignacio Marín-Jiménez, Luis Menchén, Emilio Laserna-Mendieta, Alfredo J. Lucendo, María Sanjurjo-Sáez, Luis A. López-Fernández

**Affiliations:** 1Instituto de Investigación Sanitaria Gregorio Marañón, Hospital General Universitario Gregorio Marañón, 28007 Madrid, Spain; 2Hospital Universitario Infantil Niño Jesús, 28009 Madrid, Spain; 3Hospital Vall d’Hebrón, 08035 Barcelona, Spain; 4Hospital Universitario A Coruña, 15006 A Coruña, Spain; 5Hospital Universitario Infanta Leonor, 28031 Madrid, Spain; 6Hospital Universitario Parc Taulí, 08208 Sabadell, Spain; 7Hospital Regional Universitario de Málaga, 29011 Málaga, Spain; 8Hospital Universitario Virgen de Valme, Universidad de Sevilla, 41014 Sevilla, Spain; 9Hospital Universitario la Paz, 28046 Madrid, Spain; 10Complejo Asistencial Universitario de Salamanca, 37007 Salamanca, Spain; 11Hospital Universitario Miguel Servet, 50009 Zaragoza, Spain; 12Servicio de Gastroenterología, Hepatología y Nutrición Pediátrica, Hospital Sant Joan de Dèu, 08950 Barcelona, Spain; 13Hospital Universitario Virgen de la Macarena, 41009 Sevilla, Spain; 14Hospital Universitario Virgen del Rocio, 41013 Sevilla, Spain; 15Hospital Universitario Doce de Octubre, 28041 Madrid, Spain; 16Departamento de Medicina, Facultad de Medicina, Universidad Complutense, 28040 Madrid, Spain; 17Departamento of Gastroenterología, Hospital General de Tomelloso, Centro de Investigación Biomédica en Red de Enfermedades Hepáticas y Digestivas (CIBERehd), Instituto de Investigación Sanitaria de Castilla-La Mancha (IDISCAM), 28029 Madrid, Spain; 18Instituto de Investigación Sanitaria La Princesa, 28006 Madrid, Spain

**Keywords:** pharmacogenetics, single nucleotide polymorphism, infliximab, adalimumab

## Abstract

The genetic polymorphisms rs2395185 and rs2097432 in HLA genes have been associated with the response to anti-TNF treatment in inflammatory bowel disease (IBD). The aim was to analyze the association between these variants and the long-term response to anti-TNF drugs in pediatric IBD. We performed an observational, multicenter, ambispective study in which we selected 340 IBD patients under 18 years of age diagnosed with IBD and treated with anti-TNF drugs from a network of Spanish hospitals. Genotypes and failure of anti-TNF drugs were analyzed using Kaplan-Meier curves and Cox logistic regression. The homozygous G allele of rs2395185 and the C allele of rs2097432 were associated with impaired long-term response to anti-TNF drugs in children with IBD after 3 and 9 years of follow-up. Being a carrier of both polymorphisms increased the risk of anti-TNF failure. The SNP rs2395185 but not rs2097432 was associated with response to infliximab in adults with CD treated with infliximab but not in children after 3 or 9 years of follow-up. Conclusions: SNPs rs2395185 and rs2097432 were associated with a long-term response to anti-TNFs in IBD in Spanish children. Differences between adults and children were observed in patients diagnosed with CD and treated with infliximab.

## 1. Introduction

Inflammatory bowel disease (IBD) comprises a group of immune-mediated diseases (Crohn’s disease (CD), ulcerative colitis (UC), indeterminate colitis (IC)) that debut in childhood in approximately 25% of cases [1]. Nevertheless, most research on this illness has been performed in adults. The aim of IBD treatment is to control symptoms and promote mucosal or even transmural healing, as assessed using imaging techniques and non-invasive biomarkers. Treatment includes aminosalicylates, corticosteroids, and immunomodulatory biological drugs [2]. One of the differences between pediatric and adult IBD is that biological drugs are used earlier in children [3,4]. The anti-tumor necrosis factor (anti-TNF) drugs infliximab (IFX) and adalimumab (ADL) are the only biological drugs approved for pediatric IBD. However, the response to these drugs is not universal: 2–9% of patients do not achieve a primary response, and 15–25% fail to achieve clinical remission after 1 year of treatment [5]. 

Most studies aim to identify patients at risk of failure of anti-TNF therapy. Clinical, biochemical, and genomic variations have been reported to be associated with the response to anti-TNF drugs [6]. Serum anti-TNF trough levels, antidrug antibody generation, fecal calprotectin, age at diagnosis, and type of IBD are associated with the response to biological drugs [7,8,9]. 

Identifying patients who are highly likely to achieve a long-term response may be extremely relevant for the treating physician. Unlike other markers, genetic variants could be very useful for predicting which patients will respond to therapy before initiation. Several single-nucleotide variants can predict the response to anti-TNF agents in children with IBD [4]. However, evidence in favor of the use of these biomarkers in clinical practice remains insufficient.

Major histocompatibility complex proteins are essential in the activation of the immune system and are encoded by the human leukocyte antigen (HLA), a group of more than 200 genes located closely together on the short arm of chromosome 6. DNA variants in HLA genes have also been associated with the response to anti-TNF agents in other inflammatory diseases, such as psoriasis [10,11] and rheumatoid arthritis [12]. In IBD, 2 variants in HLA genes (rs2097432 and rs2395185) have been associated with response to anti-TNF drugs. The SNP rs2097432 C, which initially correlated with HLA-DQA1*05, predicts anti-IFX antibody formation and loss of response in adults with IBD [13], and antidrug antibodies to IFX and ADL in patients aged 6 years or older with CD [14]. Antidrug antibody formation is one of the most relevant laboratory parameters driving switches in biological treatment. Despite this limited evidence, rs2097432 is beginning to be genotyped in clinical practice in several hospitals for both adults and children diagnosed with IBD. The SNP rs2395185 G, an intronic variant in HLA-DRB9, was also associated with the risk of IBD and primary non-response to IFX in a small group of patients with IBD and with an age at diagnosis < 21 years [15]. 

The objectives of this study were to explore the association between 2 SNPs in MHC class II, rs2395185 and rs2097432, and long-term response to anti-TNF drugs in children with IBD and compare this association with a Spanish adult population of CD patients.

## 2. Results

### 2.1. Patients

A total of 340 children diagnosed with IBD were recruited and followed for up to 9 years (Table 1). Most patients were male (60.3%), had CD (70.5%), were biological drug-naïve (93.5%), and had received IFX (67.1%). In addition, most were in combination therapy with an immunosuppressant (86.7%). At the end of follow-up, anti-TNF therapy failed in 70 patients. Comparison of patients whose anti-TNF treatment failed with those whose therapy did not reveal statistically significant differences between both populations for type of IBD and line of biological treatment (*p* value 0.012 and 0.001, respectively).

A population of adults diagnosed with CD and treated with IFX was used to compare the results obtained with these 2 SNPs in children (Table 2). The cohort was followed for up to 9 years. The only statistically significant difference between the failure and non-failure of anti-TNF therapy in adults was for combination therapy. 

### 2.2. Genotyping and Frequencies 

Genotyping was successful in both rs2395185 and rs2097432, although genotyping of rs2097432 using a TaqMan probe was unsuccessful in 17 children. Subsequent Sanger sequencing classified these patients as homozygous wild type. 

Hardy-Weinberg equilibrium (HWE) was analyzed for deviations in genotype frequency. The observed genotype frequencies in rs2097432 were not consistent with HWE. The frequency of the reference G allele for SNP rs2395185 was 0.69, while the frequency of the reference T allele for rs2097432 was 0.67. These frequencies were consistent with those observed in the Genome Aggregated Database (gnomAD) in the most similar population (European non-Finnish) (0.7101 and 0.6262, respectively) [16]. 

The linkage disequilibrium study showed that the heritability of both SNPs was high (D’ = 0.9443). However, the degree of correlation was low (R^2^ = 0.1482).

### 2.3. Association of rs2097432 with Response to Anti-TNF Drugs

The rs2097432 C allele was associated with the long-term failure of anti-TNF drugs in a univariate analysis. Carriers of the rs2097432 C allele had a greater risk of failure during the 9 years of follow-up (*p* value 0.028) (Figure 1a). This association was maintained after a multivariate logistic regression analysis (hazard ratio (HR), 1.770; 95% CI, 1.100–2.848; *p* value 0.019), where type of IBD and type of anti-TNF drug were also independent variables associated with response (HR, 2.235; 95% CI, 1.363–3.666; *p* value 0.001; and HR, 1.7; 95% CI, 1.018–2.777; *p* value 0.043, respectively). A non-statistically significant trend was also observed after a follow-up to 3 years (*p* value 0.058) (Figure 1b). This became statistically significant after a multivariate logistic regression analysis (HR 1.693; 95% CI, 1.014–2.825; *p* value 0.044). In addition, the time to failure was shorter for children carrying the C allele than for those who did not (9.13 versus 13.7 months).

Since the type of IBD and type of anti-TNF were independently associated with the long-term response to anti-TNF therapy, new analyses using Kaplan-Meier curves were performed for children diagnosed with CD or UC and those treated with IFX or ADL. When the cohort of children was separated by type of IBD, rs2097432 maintained its association with long-term response in those diagnosed with CD (*p* value 0.026) (Figure 1c), but not in those diagnosed with UC (*p* value 0.353) (Figure 1d). This association was maintained after multivariate analysis (HR, 2.068; 95% CI, 1.071–3.994; *p* value 0.030).

When the cohort was analyzed based on the anti-TNF drug used, the association between this SNP and the response to the anti-TNF drug was lost in children treated with IFX and in those treated with ADL (*p* value 0.252 and *p* value 0.055, respectively) (Figure 1e,f). However, after multivariate analysis, the association of ADL with response became statistically significant (HR, 2.319; 95% CI, 1.022–5.260; *p* value 0.044). This trend was not evident during the first 2 years of treatment.

### 2.4. Association between rs2395185 and Response to Anti-TNF Drugs

The univariate analysis showed the rs2395185 G allele to be associated with long-term failure of anti-TNF drugs. Carriers of the T rs2395185 allele were at a lower risk of failure after a follow-up of 9 years (*p* value 0.035) (Figure 2a). This association was maintained in a multivariate logistic regression analysis (HR, 0.599; 95% CI, 0.368–0.975; *p* value 0.039), both for type of IBD (CD reference vs. UC), and anti-TNF treatment (IFX reference vs. ADL) (HR, 2.070; 95% CI, 1.269–3.376; *p* value 0.004 and HR, 1.742; CI 95%, 1.057–2.872; *p* value 0.030, respectively). This association was also observed after a follow-up to 3 years (*p* value 0.044) (Figure 2b), although statistically, significance was lost in the multivariate logistic regression analysis (HR 0.603; 95% CI, 0.355–1.024; *p* value 0.061). In addition, the time to failure was larger for children carrying the T allele than for those who did not (16.0 versus 9.16 months). 

When the cohort of children was separated by type of IBD, rs2395185 lost the association with long-term response in those diagnosed with CD (Figure 2c), those with UC (Figure 2d), and those treated with IFX or ADL (Figure 2e,f). However, in all cases, a trend toward loss of response with anti-TNFs was observed in GG patients.

### 2.5. Haplotypes of rs2395185 and rs2097432 with Response to Anti-TNF Drugs

The putative effect on the anti-TNF drug response of the haplotypes of these 2 SNPs was also studied. Since rs2097432 CC or CT and rs2395185 GG are risk genotypes for anti-TNF failure, we defined three risk groups by haplotype (Table 3). 

Clinical characteristics of pediatric patients classified according to their haplotypes and risk of failure of anti-TNF drug showed, as expected, a higher percentage of failure and a shorter time from treatment initiation to failure in high-risk haplotypes, although the last variable was not statistically significant (Appendix A).

The univariate analysis showed a statistically significant association with risk of failure when comparing low-risk versus medium- or high-risk groups (*p* value 0.022) (Figure 3a). This association was significant after multiple Cox multivariate regression (HR, 2.079; 95% CI, 1.170–3.694; *p* value 0.013).

The association became more significant with the higher risk groups we considered. Lower rates of failure were observed in patients with low-risk haplotypes than in medium-risk or high-risk haplotypes (*p* value 0.006) (Figure 3b). This association was also significant after multiple Cox multivariate regression (HR, 2.409; 95% CI, 1.282–4.527; *p* value 0.006).

### 2.6. Comparison of Children with Adults with Crohn’s Disease and Treated with IFX

Our cohort of adults diagnosed with CD enabled us to analyze and compare the association between rs2097432 (Figure 4) and rs2395185 (Figure 5) using Kaplan-Meier curves and the long-term response to IFX in adults and children with CD. The SNP rs2097432 was not associated with failure of IFX in adults with CD when the follow-up period was 9 years (*p* value 0.246), or when it was 3 years (*p* value 0.060). However, after a Cox multivariate regression analysis, the difference for rs2097432 was close to statistical significance after 3 years of follow-up (HR, 2.164; CI 95%, 0.996–4.701; *p* value 0.051;). In addition, the SNP rs2395185 was associated with failure of IFX in adults with CD after 3 or 9 years of follow-up (*p* value 0.015 and *p* value 0.032, respectively). The associations were maintained in a multivariate analysis after 3 years (HR, 0.450; CI 95%, 0.206–0.982; *p* value 0.045) and 9 years (HR, 0.523; CI 95%, 0.303–0.902; *p* value 0.020). Neither of these SNPs was associated with this variable in children with CD.

## 3. Discussion

Identifying which patients with IBD are at risk of failure of anti-TNF drugs and which have a high probability of maintaining a response to anti-TNF drugs over a long period would help considerably in personalizing treatments and avoiding the use of immunosuppressants in individuals for whom they are not necessary. Combination therapy comprising anti-TNFs and immunosuppressants reduces immunogenicity but increases the risk of infections and hematological toxicity. For this reason, identifying predictive biomarkers of anti-TNF responses may be useful. Several SNPs in TNF and TNF receptors and in apoptosis and autophagy genes have been involved in the response to anti-TNF drugs in patients with IBD [17].

Antidrug antibody formation is a relevant cause of the loss of response in patients treated with biological drugs in IBD [18]. Some factors, such as smoking, infections, antibiotics, and immunosuppressants, are related to antidrug antibody formation in autoimmune diseases [19]. However, genetic factors have a very important role in antidrug antibody formation [19], with HLA haplotypes being one of the most promising for predicting patients at risk of immunogenicity [20]. The MHC region contains more than 250 genes (including some HLA genes among them), and linkage disequilibrium blocks of up to 3 Mb are frequent. 

In this study, we confirm the association of 2 genetic variants in the HLA region, rs2097432 and rs2395185, with failure of anti-TNF drugs in children. In addition, we demonstrate differences in the characteristics of these associations in children and adults, and over time. 

Both SNPs were associated with failure and long-term effectiveness of anti-TNF drugs in children with IBD during a follow-up period ranging from 3 to 9 years. Most published studies look for primary nonresponse or loss of response after short follow-up periods [13,14,21,22,23]. Our strategy shows how responses progress over long periods of time.

The intronic variant rs2097432 is located on chromosome 6 at position 32590771 T>C (GRCh37) near *HLA-DQA1*. Our results showed that the C allele of this SNP was associated with a higher risk of failure of anti-TNF drugs in children followed for up to 9 years. This trend was also observed after 3 years of follow-up in a multivariate analysis. The PANTS study showed rs2097432 (*HLA-DQA1**05) to be associated with the development of antidrug antibodies to IFX and ADL in patients aged ≥6 with CD followed over 200 months [14]. Wilson et al. found that the rs2097432 variant predicted antidrug antibody formation to IFX or ADL therapy for IBD in adults [13]. The authors found that up to 79% of patients with anti-IFX antibodies were carriers of at least 1 C allele. However, the patients recruited in these studies were mainly adults, leaving the situation of children largely unaddressed. A recent multicenter study of children with CD treated with IFX followed for 1 year for the formation of antidrug antibodies revealed that a neutrophil CD64 ratio >6 and starting dose <7.5 mg/kg independently predicted antidrug antibody formation, while *HLA-DQA1**05 did not [21]. The resolution of 2 digits in *HLA-DQA1**05 may be insufficient to discriminate between patients at risk or not of developing immunogenicity. A study discriminating 4 digits using PANTS cohort data showed that *DRB1**03:01, *DQA1**05:01, and *DQB1**02:01 were associated with IFX antibody formation, while *DRB1**11:01, *DQA1**05:05, and *DQB1**03:01 were associated with ADL antibody formation [22]. However, these conclusions are controversial, and larger cohorts will be required to test specific DQA1∗05 alleles for an association with ADL and IFX individually [23]. Our results showed a trend toward significance for a higher risk of failure to ADL in carriers of the C allele of rs2097432. No differences were observed for IFX-treated children, and while this observation does not help to resolve the controversy, since we analyzed only 1 SNP, it does reflect differences regarding this SNP in IFX- or ADL-treated patients. In this sense, we identified genetic variants in *TLR4*, *LY96*, *TLR2*, *CD14*, and *TNFRSF1B* that were differentially associated with IFX or ADL though serum levels [24]. The evidence suggests that genetic variants might play a role in the differential effectiveness of IFX or ADL in IBD. 

The intronic variant rs2395185 is located on chromosome 6 in position 324633167 (GRCh37) in a region spanning *BTNL2* to *HLA-DQB1*, and it has been described in the literature on *HLA-DRA* and *HLA-DRB1*. However, it is currently considered part of the *HLA-DRB9* gene. Our findings showed a higher risk of failure of anti-TNF drugs in homozygous GG children than in GT or TT children with IBD. Thus, it is observed after 3 and 9 years of starting biological treatment in children with IBD. It was first associated with the risk of UC by GWAS (*HLA-DQB1*) [25,26] and was further analyzed in an Italian population [27]. Carriers of the T allele have a lower risk of UC: 3.7% vs. 5.1% in homozygous TT in UC vs. control, 28.1% vs. 35.7% in heterozygous TG in UC vs. control, and 68.2% vs. 59.2% in homozygous GG in UC vs. control [27]. In a small group of patients, Dubinsky et al. showed the G allele of this SNP to be associated with the risk of IBD, primary non-response to IFX in patients with IBD, and age at diagnosis <21 years [15]. Thus, our results extend the association between this SNP and long-term response to anti-TNF agents and help identify children who will respond well to anti-TNF drugs over a long period.

Comparison of children and adults was limited to CD and IFX as treatments and revealed differences between adults and children. Thus, rs2395185 was associated with the response to IFX in adults diagnosed with CD, but not in children with CD after follow-ups of 3 or 9 years. Differences between adults and children with UC in rs2395185 have already been defined as follows: in an Italian cohort, 3.1% in adults with UC and TT genotype vs. 6% in children; 28.4% in adults with UC and TG genotype vs. 26.9% in children; and 68.5% in adults with UC and GG genotype vs. 67.1% in children (HR 0.67; 95% CI, 0.52–0-86; *p* value 0.002) [27]. Our results suggest small differences depending on the anti-TNF drug, type of IBD, and, probably, patient age.

Although our cohort of children with IBD is one of the largest in the world, our cohort of adults is limited because only CD and IFX can be compared. We are now recruiting a large number of adult patients with IBD treated with biological drugs for future studies. Another limitation is that only IFX and ADL were assessed, while both children and adults are increasingly treated with other biological drugs, such as vedolizumab and ustekinumab. It would be interesting to study the role of these SNPs in both treatments. However, these drugs are rarely used in children, thus making sample size a considerable limitation. 

The genetic variant rs2097432 was not associated with the time to failure of IFX in adults or children with CD. However, there was an almost statistically significant trend after 3 years of follow-up in the adult cohort. Our results show that the follow-up period is extremely sensitive for finding associations. Most studies attempting to identify genetic biomarkers of response to biological drugs analyzed primary or short-term response (less than 1 year) and established a cut-off point. However, our strategy allows us to measure how the association between the marker and the response progresses over long periods and provides us with valuable information to identify patients whose prognosis is very good with biological drugs. In patients who have lived with this incurable illness for 50–70 years, the selection of the most durable and efficient treatment is highly relevant. 

The linkage disequilibrium of HLA genes is well known. An analysis of these two SNPs in the LDpair tool in the population most similar to the Spanish one showed a high common heritability but a low correlation. These data justify the study of these variants independently. Although genotype frequencies in rs2097432 were not consistent with HWE, the frequency of the C allele is similar to the frequency expected by genomic databases, such as gnomAD, and the detection of carriers of the C allele for rs2097432 is 94.9% concordant with diagnostic methods for detecting carriers of *HLA-DQA1*:05 [28], we consider our results to be robust, even if the Hardy-Weinberg equilibrium is not supported.

Some authors recommend testing *HLA-DQA1**05:01, *HLA-DQA1**05:05, and *HLA-DRB1**03 to predict immunogenicity to anti-TNF drugs, with the aim of choosing therapy and preventive strategies [29]. Other researchers have also included rs2097432 in clinical trials to study how it can be applied in clinical practice (INHERIT, https://ichgcp.net/clinical-trials-registry/NCT04109300#; accessed on 10 January 2023). Since rs2097432 genotyping is being introduced in trials for implementation in clinical practice, we suggest the inclusion of rs2395185 in clinical trials, at least in Caucasian patients. Such an approach would enable us to personalize biological therapy in children based on the individual genome.

## 4. Materials and Methods

### 4.1. Study Design and Patient Characteristics 

This study was observational, ambispective, and longitudinal. Participants were informed about the study and recruited between 2016 and 2022 in the pediatric departments of various Spanish hospitals after signing the informed consent document. Patients (*n* = 340) were recruited based on the following inclusion criteria: current diagnosis of IBD, age under 18, and having received treatment with an anti-TNF drug at any time. Participants were followed up until March 2022. Data were collected from clinical records and managed using REDCap (Research Electronic Data Capture) electronic data capture tools hosted at Hospital General Universitario Gregorio Marañón [30,31]. Samples from patients diagnosed with CD but aged ≥ 18 were used for comparison [3]. These samples were obtained from the registered collection c.0003459 (Instituto de Salud Carlos III), according to Spanish legislation. The characteristics of the adult participants were updated from Salvador-Martín et al. [3]. 

Time-to-failure was calculated as the time from the start of therapy to the date of failure and modification of treatment. Failure of anti-TNF therapy was defined as withdrawal of the anti-TNF drug, and/or switching to another drug due to loss of effectiveness according to clinical, biochemical, and endoscopic data or the need for abdominal surgery related to progression of IBD.

The other clinical and demographic variables collected were sex, age at onset of IBD, time from diagnosis to initiation of the anti-TNF drug, type of IBD, and type of anti-TNF drug. Type of anti-TNF drug and type of IBD were analyzed as potential sources of bias.

The sample size was calculated as per Salvador-Martín et al. [3].

### 4.2. DNA Isolation and Genotyping 

DNA was isolated from 200 µL of blood using the QIAmp DNA Blood mini kit (Qiagen, Hilden, Germany), quantified in a Quawell 5000 spectrophotometer (Quawell Technology Inc., San José, CA, USA), and diluted to 10 ng/µL for genotyping. Genotyping was performed using the QuantStudio 3 Real-Time PCR System (Applied Biosystems, Carlsbad, CA, USA). rs2097432 was genotyped using a rhAMP probe (Integrated DNA Technologies, Coralville, IA, USA), while rs2395185 was genotyped using a TaqMan Probe (Applied Biosystems). Both genotypes were obtained following the manufacturer’s recommendations. For samples with no real-time genotyping result for rs2097432, genotyping was performed using Sanger sequencing. The fragment was amplified by mixing 10 ng genomic DNA, 1 µM of each primer (forward 5′-ACATCCTGTGACCAAACAGCT-3′, reverse 5′-TGTGCACCTACCCTCACT-3′), and 5 µL of Complex Master Mix Taq polymerase in a final volume of 10 µL. PCR conditions were as follows: initial denaturation at 95 °C for 2 min; 35 cycles of 95 °C for 30 s, 60 °C for 30 s, and 72 °C for 90 s; and a final extension at 72 °C for 5 min. Genotypes were analyzed using QuantStudio^TM^ Design & Analysis Software (Applied Biosystems). Linkage disequilibrium between rs2097432 and 2395185 was determined using the LDpair tool (National Cancer Institute, https://analysistools.cancer.gov/LDlink/?tab=ldpair; accessed on 10 January 2023).

### 4.3. Statistical Analysis 

Quantitative continuous variables were expressed as median and interquartile range (IQR), whereas qualitative variables were expressed as frequency and percent. 

The statistical analysis was performed using IBM SPSS Statistics v.26 (IBM Corp., Armonk, NY, USA). Differences between qualitative and quantitative variables were tested using the chi squared test, or, whenever appropriate, the Fisher exact test or Mann–Whitney test, respectively. Differences between quantitative and qualitative variables in more than two independent groups were analyzed using the Kruskal-Wallis test.

Kaplan-Meier curves were used to analyze the association between genotypes and the long-term response to anti-TNF drugs. For SNPs with a *p* value < 0.05, the adjusted hazard ratios (HR) and their 95% confidence interval (CI) were calculated using Cox regression based on sex, anti-TNF treatment, and type of IBD as covariates. A *p* value of less than 0.05 was considered statistically significant. 

### 4.4. Ethics 

The study was approved by the ethics committees of the participating hospitals and conducted in accordance with the World Medical Association Declaration of Helsinki and Spanish legislation. All patients or legal guardians gave their signed informed consent for genetic testing and participation in the study. 

## 5. Conclusions

The genetic variants rs2097432 and rs2395185 were associated with a long-term response to anti-TNF drugs in Spanish children with IBD. An additive effect of both risk alleles was observed. There were differences in the anti-TNF response between adults and children with CD treated with IFX, mainly for the genetic variant rs2395185. Genotyping of these SNPs could help clinicians personalize treatments. 

## Figures and Tables

**Figure 1 ijms-24-01797-f001:**
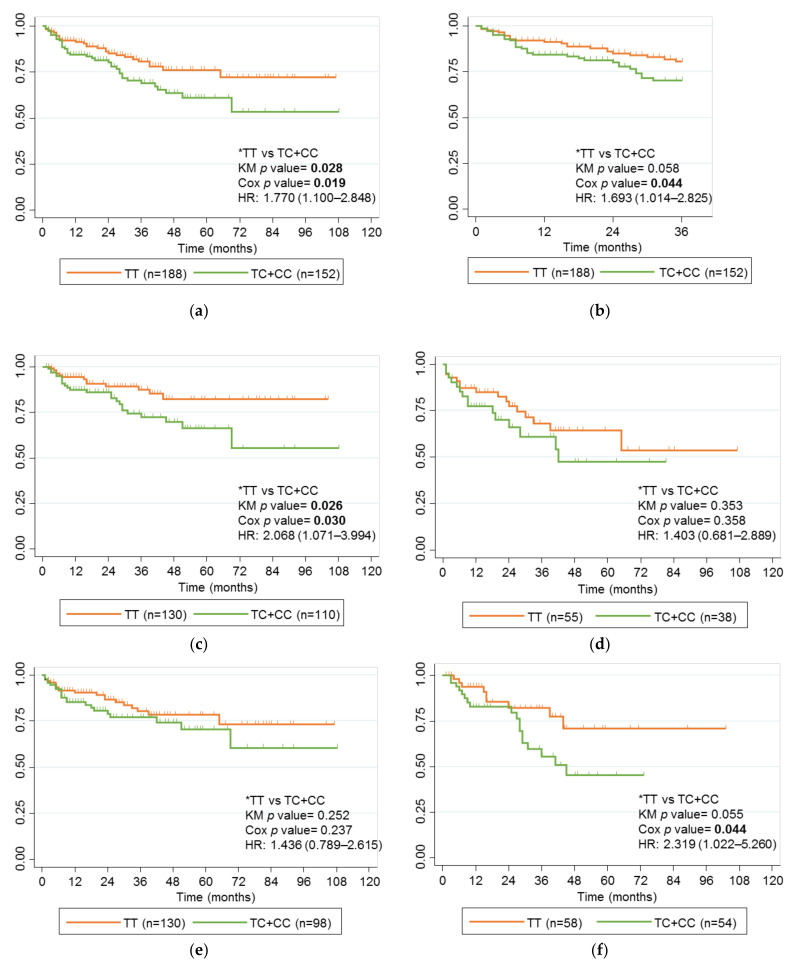
Kaplan-Meier curves for rs2097432 and failure of anti-TNF therapy. (**a**) Children with IBD treated with anti-TNF drugs; (**b**) Children with IBD treated with anti-TNF drugs for 36 months; (**c**) Children with CD treated with anti-TNF drugs; (**d**) Children with UC treated with anti-TNF drugs; (**e**) Children with IBD treated with IFX; (**f**) Children with IBD treated with ADL. Genotype comparisons and *p* values for the univariate analysis (KM *p* value) and multivariate analysis (Cox *p* value) analyses are presented alongside Kaplan-Meier curves. Significant *p* values are marked in bold. * Reference genotypes.

**Figure 2 ijms-24-01797-f002:**
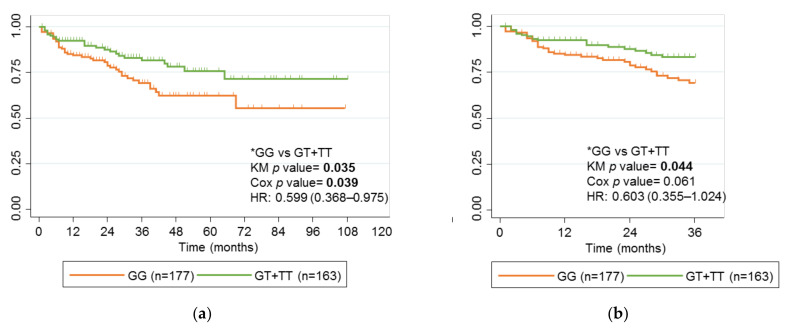
Kaplan-Meier curves for rs2395185 and failure of anti-TNF therapy. (**a**) Children with IBD treated with anti-TNF drugs; (**b**) Children with IBD treated with anti-TNF drugs for 36 months; (**c**) Children with CD treated with anti-TNF drugs; (**d**) Children with UC treated with anti-TNF drugs; (**e**) Children with IBD treated with IFX; (**f**) Children with IBD treated with ADL. Genotype comparisons and *p* values for univariate analysis (KM *p* value) and multivariate analysis (Cox *p* value) are presented alongside Kaplan-Meier curves. Significant *p* values are marked in bold. * Reference genotypes.

**Figure 3 ijms-24-01797-f003:**
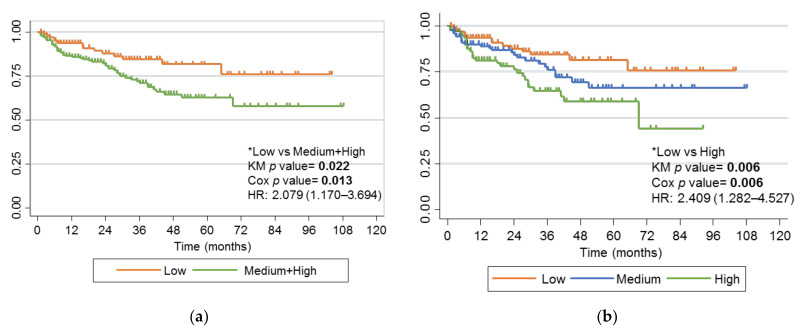
Kaplan-Meier curve of combined effect of rs2395185 and rs2097432 on failure of anti-TNF drugs. (**a**) Patients carrying a low-risk haplotype (orange) or any other haplotypes (green); (**b**) Patients carrying a low-risk haplotype (orange), a medium-risk haplotype (blue), or a high-risk haplotype (green). Genotype comparisons and *p* values for the univariate analysis (KM *p* value) and multivariate analysis (Cox *p* value) analyses are presented alongside Kaplan-Meier curves. Significant *p* values are marked in bold. * Reference genotypes.

**Figure 4 ijms-24-01797-f004:**
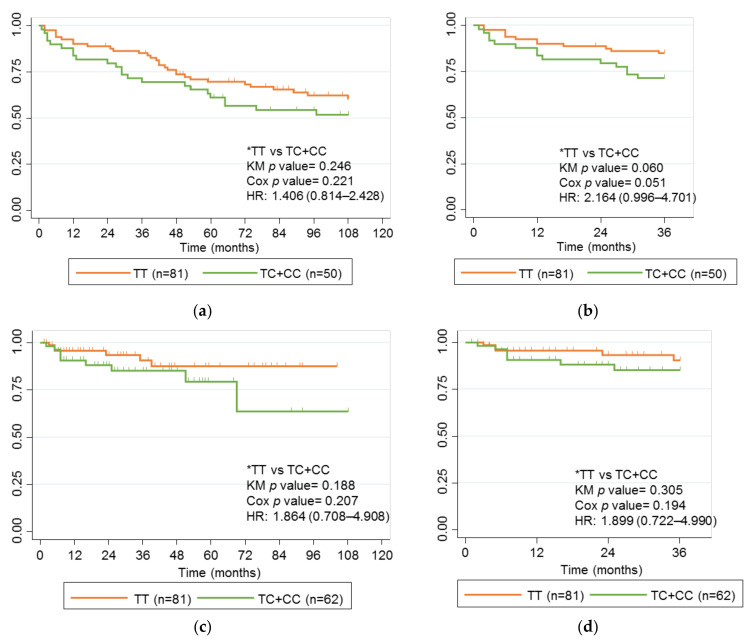
Kaplan-Meier curves for rs2097432 and failure of IFX in children and adults with CD. (**a**) Adults with CD treated with IFX, follow-up 9 years; (**b**) Adults with CD treated with IFX, follow-up 3 years; (**c**) Children with CD treated with IFX, follow-up 9 years; (**d**) Children with CD treated with IFX, follow-up 3 years. Genotype comparisons and *p* values for the univariate analysis (KM *p* value) and multivariate analysis (Cox *p* value) analyses are presented alongside Kaplan-Meier curves. Significant *p* values are marked in bold. * Reference genotypes.

**Figure 5 ijms-24-01797-f005:**
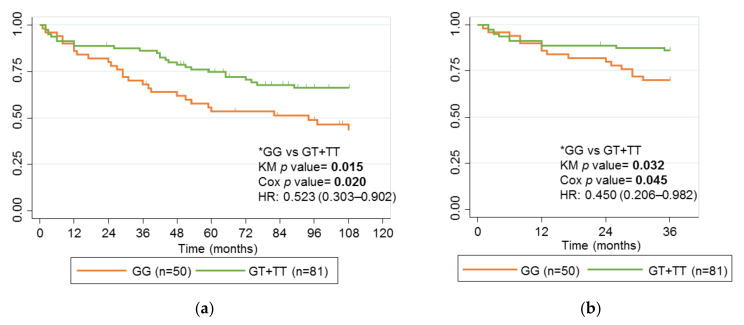
Kaplan-Meier curves for rs2395185 and failure of IFX in children and adults with CD. (**a**) Adults with CD treated with IFX, follow-up 9 years; (**b**) Adults with CD treated with IFX, follow-up 3 years; (**c**) Children with CD treated with IFX, follow-up 9 years; (**d**) Children with CD treated with IFX, follow-up 3 years. Genotype comparisons and *p* values for the univariate analysis (KM *p* value) and multivariate analysis (Cox *p* value) analyses are presented alongside Kaplan-Meier curves. Significant *p* values are marked in bold. * Reference genotypes.

**Table 1 ijms-24-01797-t001:** Characteristics of pediatric patients.

Characteristics	Overall (*n* = 340)	Responders (*n* = 270)	Non-Responders (*n* = 70)	*p* Value
Sex				
Male, *n* (%)	205 (60.3%)	168 (82.0%)	37 (18.0%)	0.171
Female, *n* (%)	135 (39.7%)	102 (75.6%)	33 (24.4%)	
Age (years)				
At diagnosis, median (IQR, range)	11.2 (4.0, 0.7–17.3)	11.2 (3.8, 0.9–17.3)	11.0 (5.8, 0.7–16.0)	0.109
At start of treatment, median (IQR, range)	12.2 (4.1, 1.1–17.5)	12.3 (4.1, 1.4–17.5)	12.0 (4.3, 1.1–17.3)	0.251
Months from diagnosis to initiation of therapy, median (IQR, range)	6.1 (15.9, 0.0–129.6)	6.1 (15.3, 0–129.6)	8.9 (18.2, 0.0–125.8)	0.202
Type of IBD				
CD, *n* (%)	240 (70.5%)	201 (83.8%)	39 (16.3%)	
UC, *n* (%)	93 (27.4%)	63 (67.7%)	30 (32.3%)	**0.012**
IC, *n* (%)	7 (2.1%)	6 (85.7%)	1 (14.3%)	
First or second biological treatment				
1st, *n* (%)	318 (93.5%)	259 (81.4%)	59 (18.6%)	**0.001**
2nd, *n* (%)	22 (6.5%)	11 (50.0%)	11 (50.0%)	
Drug				
Infliximab, *n* (%)	228 (67.1%)	185 (81.1%)	43 (18.9%)	0.318
Adalimumab, *n* (%)	112 (32.9%)	85 (75.9%)	27 (24.1%)	
Concomitant immunomodulator (*n* = 330)				
Yes, *n* (%)	286 (86.7%)	229 (80.1%)	57 (19.9%)	0.843
No, *n* (%)	44/13.3%)	36 (81.8%)	8 (18.2%)	0.843
Type of immunotherapy (*n* = 283)				
Azathioprine	266 (81.6%)	215 (80.8%)	51 (19.2%)	
Mercaptopurine	3 (0.9%)	2 (66.7%)	1 (33.3%)	0.947
Methotrexate	14 (4.3%)	10 (71.4%)	4 (28.6%)	

Bold, *p* value < 0.05; IQR, Interquartile range; CD, Crohn’s disease; UC, Ulcerative colitis; IC, indeterminate colitis.

**Table 2 ijms-24-01797-t002:** Characteristics of adult patients.

Characteristics	Overall (*n* = 131)	Responders (*n* = 77)	Non-Responders (*n* = 54)	*p* Value
Sex				
Male, *n* (%)	66 (50.4%)	41 (62.1%)	25 (37.9%)	0.480
Female, *n* (%)	65 (49.6%)	36 (55.4%)	29 (44.6%)	
Age (years)				
At diagnosis, median (IQR, range)	27.2 (16.4; 10.8–76.7)	27.7 (18.2; 10.8–76.7)	26.7 (15.9; 11.7–58.7)	0.359
At start of treatment, median (IQR, range)	37.4 (18.5; 12.5–81.4)	36.6 (19; 12.5–81.4)	35.5 (18.2; 16.2–64.5)	0.818
First or second biological treatment				
1st, *n* (%)	119 (90.8%)	73 (60.3%)	48 (39.7%)	0.317
2nd, *n* (%)	12 (9.2%)	4 (40%)	6 (60%)	
Concomitant immunomodulator (*n* = 128)				
Yes, *n* (%)	68 (51.9%)	47 (69.1%)	21 (30.9%)	**0.045**
No, *n* (%)	60 (45.8%)	30 (50%)	30 (50%)	
Type of immunotherapy				
Azathioprine	54 (79.4%)	39 (72.2%)	15 (27.8%)	0.870
Mercaptopurine	4 (5.9%)	4 (100%)	0 (0%)	
Methotrexate	7 (10.3%)	2 (28.6%)	5 (71.4%)	
Mycophenolate	3 (4.4%)	2 (66.7%)	1 (33.3%)	

Bold, *p* value < 0.05; IQR, Interquartile range.

**Table 3 ijms-24-01797-t003:** Classification of haplotype and risk of anti-TNF drug response.

rs2097432	rs2395185	Risk Group
TT	TT or GT	Low
CC ^1^ or CT ^1^	TT or GT	Medium
TT	GG ^1^	Medium
CC ^1^ or CT ^1^	GG ^1^	High

^1^ Risk genotype.

## Data Availability

Data available under request.

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
