# Peer review of "Association between HLA DNA Variants and Long-Term Response to Anti-TNF Drugs in a Spanish Pediatric Inflammatory Bowel Disease Cohort"

_ijms, 2023, doi:10.3390/ijms24021797_

Round 1
Reviewer 1 Report
The authors explore the role of 2 genetic variants in HLA region (rs2097432 and rs2395185), in anti-TNF drugs response, in children suffering from IBD. This study allowed to measure how the association between the marker and the response progress over long periods and provided valuable information to identify patients whose prognosis is very good with biological drugs.
The manuscript would be improved before publication with these minor clarifications:
· - Are the 2 genetic variants analysed in Hardy-Weinberg equilibrium?
· - It is known that the concomitant treatment with immunosuppressants reduces immunogenicity. Have authors assessed by a multivariate analysis if this variable influences the long-term response in the cohort analysed?

Author Response
Thanks for all suggestions

Reviewer 2 Report
This study confirms the previously reported relationship between two SNPs in the HLA region and the treatment retention rate of anti-TNFα antibodies, mainly in pediatric patients with inflammatory bowel disease. The results are very interesting and are important findings for future personalized medicine.
Although few, the following points should be clarified
Major
1 The presence or absence of concomitant immunomodulators is an important factor in treatment retention; is it possible to show in Kaplan-Meier the change in treatment retention rate with or without concomitant IM and in combination with SNPs?
2 Please indicate in the supplemental table the number of cases and differences in clinical context for each combination of the two SNPs. Are these two SNPs in linkage disequilibrium?
Minor
3 Line 258: s2097432 -> rs2097432
4 Line 308: IDB-> What is IDB?
Author Response
Thanks for all suggestions

Round 2
Reviewer 2 Report
Thank you for your revised manuscript.